# Improving Unsupervised Relation Extraction by Augmenting Diverse Sentence Pairs

**Qing Wang, Kang Zhou, Qiao Qiao, Yuepei Li, Qi Li**

Department of Computer Science, Iowa State University, Ames, Iowa, USA

{qingwang, kangzhou, qqiao1, liyp0095, qli}@iastate.edu

## Abstract

Unsupervised relation extraction (URE) aims to extract relations between named entities from raw text without requiring manual annotations or pre-existing knowledge bases. In recent studies of URE, researchers put a notable emphasis on contrastive learning strategies for acquiring relation representations. However, these studies often overlook two important aspects: the inclusion of diverse positive pairs for contrastive learning and the exploration of appropriate loss functions. In this paper, we propose AugURE with both within-sentence pairs augmentation and augmentation through cross-sentence pairs extraction to increase the diversity of positive pairs and strengthen the discriminative power of contrastive learning. We also identify the limitation of noise-contrastive estimation (NCE) loss for relation representation learning and propose to apply margin loss for sentence pairs. Experiments on NYT-FB and TACRED datasets demonstrate that the proposed relation representation learning and a simple K-Means clustering achieves state-of-the-art performance. Source code is available[1].

## 1 Introduction

Unsupervised Relation Extraction (URE) aims to extract relations between named entities from the raw text in an unsupervised manner without relying on annotated relations from human experts. URE tasks are proposed to overcome the limitations of traditional Relation Extraction (RE) tasks, which target to extract pre-defined relations and require a time-consuming and labor-intensive data annotation process.

To discover relational facts without any pre-defined relation types, URE tasks are often formulated as semantics clustering tasks, where the entity pairs with the same relations should be clustered together. Therefore, effectively learning the relation representations is an essential step toward

solving URE tasks. Recent studies on relation representation learning (Liu et al., 2021; Zhang et al., 2021; Liu et al., 2022) adopt a contrastive learning process. Contrastive learning aims to bring similar-relation sentences (positive pairs) closer while separating different-relation sentences (negative pairs). There are several ideas for constructing positive pairs, a key step in the contrastive learning process. Some methods propose to augment the entity pairs and relation context of the original sentences as positive examples (Liu et al., 2021, 2022). To avoid similar sentences being pushed apart in the contrastive learning process, Liu et al. (2022) further propose to utilize cluster centroids to generate positive pairs within the clusters. However, these approaches often fail to provide diverse positive pairs for contrastive learning to obtain more effective relation representations.

Another challenge arises from the loss function perspective. The most recent URE work (Liu et al., 2022) directly transfers the Noise-Contrastive Estimation (NCE) loss function from computer vision tasks (He et al., 2020; Caron et al., 2020; Li et al., 2021) and achieves good performance. However, the negative examples used by the NCE loss function are randomly sampled sentences from the same minibatch, which may result in high false negatives, especially when the total number of relational clusters is not large. Moreover, relation semantics should not be regarded as "same/different" in many cases but rather as a similarity spectrum. For example, the relation *org:top_members/employees* is semantically closer to *org:members* than *org:founded_by*. Therefore, we argue that the binary NCE loss function may suit poorly for relation representation learning in URE tasks, whereas a margin-based loss may fit better.

In this paper, we propose several techniques to augment positive pairs to increase their diversity and strengthen the discriminative power of con-

trastive learning. Specifically, we propose within-sentence pairs augmentation and augmentation through cross-sentence pairs extraction. Within-sentence pairs augmentation generates positive pairs from the same sentence using intermediate words sampling and entity pair replacement techniques. In the augmentation through cross-sentence pairs extraction, we aim to recognize sentences with the same relation semantics to form a more diverse positive sample pool. Specifically, we employ a three-step approach. First, we utilize the Open Information Extraction (OpenIE) model (Angeli et al., 2015) to extract relation templates among all sentences. Intuitively, sentences with the same relation template are likely to express the same relation. To further group the templates that share the same relation semantics, we employ Natural Language Inference (NLI) (Zhuang et al., 2021) to discover mutually entailed templates. Finally, we extract positive pairs from different sentences within the same template groups. To facilitate the extraction of more cross-sentence positive pairs, we further use ChatGPT to rewrite the original sentences. To overcome the limitations of the binary NCE loss function, we propose to use margin loss to capture the spectrum of semantic similarity between positive and negative pairs.

In summary, our main contributions are:

- We propose AugURE with both within-sentence pairs augmentation and augmentation through cross-sentence pairs extraction to increase the diversity of positive pairs and enhance the discriminative power of contrastive learning for relation representations.

- We identify the limitation of NCE loss for relation representation learning and propose to apply margin loss for sentence pairs.

- Experiments demonstrate that the combination of our relation representation learning and a straightforward K-Means clustering approach achieves state-of-the-art performance on NYT-FB and TACRED datasets.

## 2 Related Work

Relation Extraction is a crucial Natural Language Processing (NLP) task and has been extensively studied in the past years (Miller et al., 1998; Zelenko et al., 2002; Zhong and Chen, 2021; Sui et al., 2023). Although existing RE methods achieve decent performance with manual annotations and KBs, those annotations can be expensive to obtain in practice. In recent years, few-shot RE and zero-shot RE (Sainz et al., 2021; Lu et al., 2022; Zhou et al., 2022) become popular, which significantly reduce the human annotation cost. However, few-shot RE and zero-shot RE still require a pre-defined relation label space which is often unavailable in new datasets or open-domain scenarios.

An upsurge of interest in URE has been witnessed to extract relational facts without pre-defined relation types and human-annotated data. The task of OpenRE has also attracted considerable attention from researchers. Unlike URE, OpenRE has additional access to existing annotated relations and aims to extract novel relational facts from open-domain corpora. Relation representation learning is a key step for both tasks. Amongst the prior studies, Tran et al. (2020) find that the utilization of named entities alone to induce relation types can surpass the performance of previous methods. This observation underscores the significant inductive bias that entity types offer for URE. Later, Hu et al. (2020) assume the cluster assignments as pseudo-labels and propose a self-supervised learning framework that leverages pre-trained language models (LMs) to learn and refine contextualized entity pair representations. However, the frequently re-initialized linear classification layers can hinder the process of representation learning. Wang et al. (2021) are the first to utilize deep metric learning in the OpenRE task and this scheme demonstrates good capabilities in representation learning.

Motivated by the achievements of contrastive learning in computer vision tasks (He et al., 2020; Caron et al., 2020; Li et al., 2021), recent state-of-the-art methods have embraced contrastive learning frameworks to acquire relation representations. Zhang et al. (2021) integrate existing hierarchical information into relation representations by a contrastive objective function. Liu et al. (2021) intervene on the named entities and context to obtain the underlying causal effects of them. Both methods adopt instance-wise contrastive learning objectives and exploit base-level sentence differences by increasing the diversity of positive examples. To further capture high-level semantics in the relation features, Liu et al. (2022) introduce a hierarchical exemplar contrastive learning schema to optimize relation representations using both instance-wise and exemplar-wise signals. However, this approach

still lacks in providing diverse positive pairs for contrastive learning, limiting the effectiveness of relation representations. Additionally, it employs the NCE loss function without considering its alignment with the task requirements.

## 3 Preliminary

### 3.1 URE Task Definition

We formalize the URE task as follows. Let $x = [x_1, ..., x_n]$ denote a sentence, where $x_i$ represents the $i$-th ($1 \leq i \leq n$) token. In the sentence, a named entity pair $(e_h, e_t)$ is recognized in advance, where $e_h = [x_{hs}, ..., x_{he}]$ represents the head entity with its start and end indices denoted as $hs$ and $he$, respectively, while $e_t = [x_{ts}, ..., x_{te}]$ represents the tail entity with its start and end indices denoted as $ts$ and $te$, respectively. Let $\mathcal{D} = [(x^1, e_h{}^1, e_t{}^1), ..., (x^N, e_h{}^N, e_t{}^N)]$ be a corpus consists of $N$ sentences along with their target entity pairs, the goal of URE is to partition the $N$ sentences into $k$ clusters, where $k$ is a predefined number of relations in the corpus.

### 3.2 Contrastive Learning

Recent studies have demonstrated the effectiveness of contrastive learning strategies to learn relation representations (Liu et al., 2021; Zhang et al., 2021; Liu et al., 2022). In these approaches, positive pairs are generally defined as two sentences with the same relation and negative pairs are two sentences with different relations. These positive and negative pairs are constructed and utilized in the contrastive objective function to bring together sentences with similar relations while separating sentences with dissimilar relations.

Liu et al. (2022) achieve remarkable success by directly adopting the Noise-Contrastive Estimation (NCE) loss function from computer vision tasks to URE's representation learning. The InfoNCE loss function (Oord et al., 2018; He et al., 2020; Li et al., 2021), which is usually used for instance-wise contrastive learning, is defined as:

$$\mathcal{L}_{InfoNCE} = \sum_{i=1}^{n} -log \frac{exp(v_i \cdot v_i'/\tau)}{\sum_{j=0}^{J} exp(v_i \cdot v_j'/\tau)}, \tag{1}$$

where $v_i$ and $v_i'$ are positive embeddings for instance $i$, $v_j'$ includes one positive embedding and $J$ negative embeddings for other instances, and $\tau$ is a temperature hyper-parameter.

## 4 Methodology

In this section, we provide a comprehensive exploration of the proposed model AugURE. Figure 1 offers an overview of our framework. As mentioned in Section 1, both positive and negative examples are important for relation representation learning in URE tasks. When handling positive examples, we proposed several augmentation techniques to generate more diverse positive pairs. When handling negative examples, we argue that relation semantics should not be simply viewed as 'same/different' but rather as a similarity spectrum. To this end, we propose to apply margin loss, which is also more robust to false negatives.

### 4.1 Relation Encoder

Given a sentence, its corresponding named entities, and entity types, the relation encoder aims to output a vector that depicts the relation between the entities. To highlight entities of interest, we decide to follow the entity marker tokens presented in Soares et al. (2019) and Xiao et al. (2020) to indicate the beginning and the end of each entity. Moreover, certain relations have been observed to exist between entities of specific types. Thus the entity type information is important guidance for the URE task and we should also incorporate the context of the entity type within each entity marker.

Specifically, for a sentence $x = [x_1, ..., e_h, ..., e_t, ..., x_n]$, we inject into it two special tokens, <e1:type> and </e1:type>, to respectively mark the beginning and the end of the head entity $e_h$, and another two special tokens, <e2:type> and </e2:type>, for the tail entity $e_t$, where "type" is replaced by the actual type of each entity. The resulting sequence of tokens becomes:

$$\widetilde{x} = [x_1, ..., \text{<e1:type>}, e_h, \text{</e1:type>}, ..., \\ \text{<e2:type>}, e_t, \text{</e2:type>}, ..., x_n]. \tag{2}$$

Considering the remarkable performance of recent deep transformers, we feed $\widetilde{x}$ into $\text{BERT}_{base}$ model made available by Devlin et al. (2019) to get the contextualized sentence encoding:

$$h = [h_1, ..., h_{\text{<e1:type>}}, h_{hs}, ..., h_{he}, h_{\text{</e1:type>}}, ..., \\ h_{\text{<e2:type>}}, h_{ts}, ..., h_{te}, h_{\text{</e2:type>}}, ..., h_n]. \tag{3}$$

Finally, to capture the relation contextual information and place more emphasis on the target entity pair, we derive the following fixed-length vector:

$$h_v = [h_{\text{<e1:type>}} | h_{\text{<e2:type>}} | h] \tag{4}$$

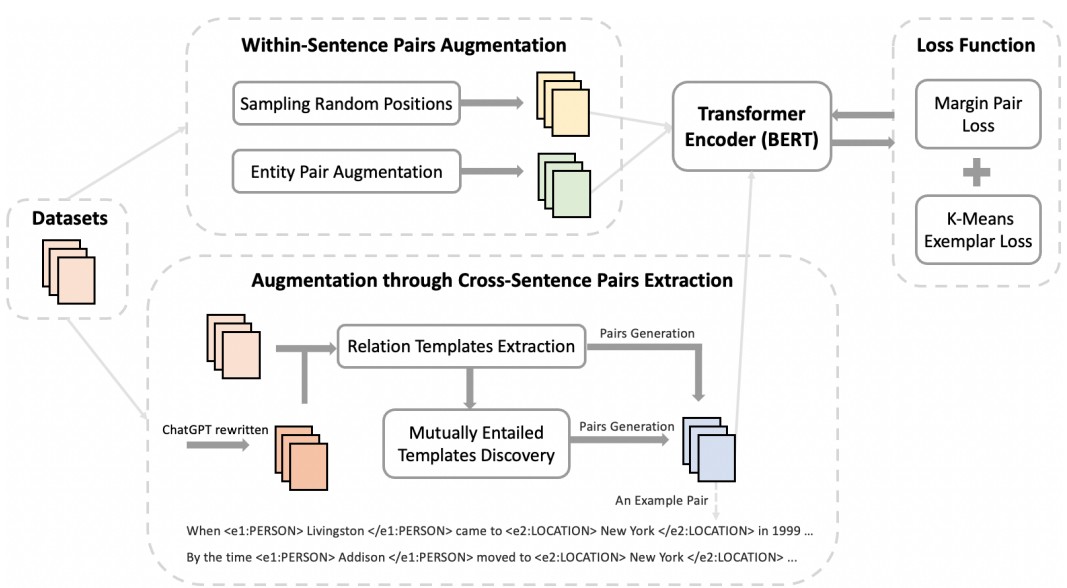

Figure 1: The framework of AugURE.

to represent the relation expressed in $x$ between the marked entities, where $a|b$ represents the concatenation of $a$ and $b$.

## 4.2 Positive Sample Augmentation

As discussed in section 1, previous methods fail to provide diverse positive pairs for contrastive learning to obtain more effective relation representations. In this section, we diversify the construction of positive pairs from two perspectives. Firstly, we augment the original sentences to generate positive examples by sampling intermediate words and replacing entity pairs. Secondly, we extract cross-sentence positive pairs by utilizing an OpenIE model to extract relation templates of the sentences and employing NLI to discover mutually entailed templates.

### 4.2.1 Within-Sentence Pairs Augmentation

Augmented positive pairs from the same sentences are important to retain the local smoothness of the contrastive learning model since the original sentence and its perturbation have similar representations. To increase the diversity of relation feature vectors generated from the same sentence $x$ and thus obtain more positive examples, we perform a random sample on the sentence encoding $h$ portion of the vector $h_v$ (Liu et al., 2022). Specifically, we use uniform sampling. The sentences are encoded first, and the sampling is conducted on the encodings of context words to achieve the variations of the context encoding. To moderately retain relation context information, during the random sample,

we only sample the non-stop words between the target entity pair (if there are less than $m$ intermediate non-stop words, sample the remaining words from other parts of the sentence). Let's denote the positions of $m$ non-duplicated randomly sampled non-stop words as $p_1, ..., p_m$, the fixed-length relation feature vector we employ is:

$$h_r = [h_{<e1:type>}|h_{<e2:type>}|h_{p_1}|...|h_{p_m}]. \quad (5)$$

By sampling $h_v$ twice, we can obtain $h_r$ and another relation feature vector

$$h'_r = [h_{<e1:type>}|h_{<e2:type>}|h_{p'_1}|...|h_{p'_m}], \quad (6)$$

where $< h_r, h'_r >$ form a positive pair.

Because $h_r$ and $h'_r$ contain the same entity pair, the representation learning may only rely on entity names and ignore the context information, limiting the representations' generalizability. To mitigate this issue and encourage the model to focus more on relation information, we propose to augment entity pairs. Because we only modify the head and tail entities, the relational context of the original sentence $x$ and the synthetic sentence $x''$ remain the same. These two sentences should have the same relation and the corresponding relation feature vectors should be a positive pair.

During the entity pair augmentation, the original $e_h$ and $e_t$ are replaced with another pair of entities $e''_h$ and $e''_t$, where $e_h$ and $e''_h$ have the same type, as do $e_t$ and $e''_t$, respectively. This synthetic sentence $x''$ derives relation feature vector $h''_r$:

$$h''_r = [h''_{<e1:type>}|h''_{<e2:type>}|h''_{p''_1}|...|h''_{p''_m}], \quad (7)$$

where $< \boldsymbol{h_r}, \boldsymbol{h_r''} >$ form a positive pair.

### 4.2.2 Augmentation through Cross-Sentence Pairs Extraction

Leveraging the original sentences and their perturbations, the learning process constructs an embedding space that primarily focuses on preserving the local smoothness around each instance while largely ignoring the diverse global information in the dataset (Li et al., 2021). Thus, relying solely on the instance-wise augmentations in section 4.2.1 is insufficient, and extracting additional high-quality cross-sentence positive pairs is essential.

We rely upon the following assumption as a basis for extracting cross-sentence positive pairs: if the target entity pairs of two sentences have the same relation textual pattern (which we call a relation template), these two entity pairs share the same underlying relation, and the corresponding relation feature vectors can be regarded as a positive pair.

To extract relation templates from the corpus without requiring any human input, we first employ an Open Information Extraction (OpenIE) (Banko et al., 2007) tool to transform the sentences into structured formats. The OpenIE model receives the raw text of the sentence as input and outputs several *(subject, predicate, object)* triples. Specifically, we utilize the Stanford OpenIE model (Angeli et al., 2015). For each sentence, we keep one valid triple if the predicate contains at least one non-stop word and the subject and object exactly match the given head and tail entities, respectively. With all valid extracted triples from the corpus, we count the frequency of distinct predicates and form high-quality relation templates to be those predicates whose frequency is greater than a threshold $t$. Finally, we enumerate the relation feature vectors of all pairs of sentences that share the same relation template to be positive pairs.

Extracting same-template sentence pairs can capture global information and diversify positive pairs, but it may be strict since different relation templates can express the same semantics meaning. For instance, "is CEO of" and "is the chief executive officer of" are two distinct relation templates but convey the same relation.

Inspired by the recent achievements of Natural Language Inference (NLI) to obtain indirect supervision in RE tasks (Sainz et al., 2021; Zhou et al., 2022), we further propose to employ an off-the-shelf NLI model (Zhuang et al., 2021) to discover templates that have the same semantic relations.

NLI, also known as Textual Entailment, aims to determine whether a "hypothesis" sentence is true (entailment), false (contradiction), or undetermined (neutral) given a "premise" sentence. In this task, we aim to discover relation templates $tp_1$ and $tp_2$, both of which are verb phrases, that are mutually entailed using NLI.

To apply NLI model, we need to construct hypothesis sentences and premise sentences using relation templates and their covered sentences, respectively. Let's denote the set of sentences whose relation template is $tp_1$ as $S_1$. To eliminate the disturbance of the specific named entities, we generate a new set $S_1'$ as premises by replacing the head and tail entities of sentences in $S_1$ with special tokens $[h]$ and $[t]$, respectively. We then synthesize a sentence $hypo_2$ in the form "$[h]$ $tp_2$ $[t]$" as a hypothesis, which is composed of the relation template $tp_2$ and the two special tokens $[h]$ and $[t]$. We call $tp_1$ entails $tp_2$ if for a certain ratio $r$ of sentences in $S_1'$ entails the synthetic sentence $hypo_2$.

NLI predicts a directional relation between two templates. To find mutually entailed relation templates, for each pair of relation templates, we conduct the above operations from both directions. Finally, the relation feature vectors of all pairs of sentences whose relation templates are mutually entailed are enumerated to be positive pairs.

The recent development of ChatGPT and its summarization power provide the possibility of capturing higher-level relations of the sentences, which can further assist in discovering more cross-sentence positive pairs. We design a simple prompt *"Given the context $\boldsymbol{x}$, what is the relationship between $\boldsymbol{e_h}$ and $\boldsymbol{e_t}$ (as short as possible)?"* to rewrite the original sentence $\boldsymbol{x}$ with the target entities $\boldsymbol{e_h}$ and $\boldsymbol{e_t}$. Given the prompt of sentence $\boldsymbol{x}$, ChatGPT will generate an output and this output is denoted as the rewritten sentence $\boldsymbol{x_c}$. Note that the rewritten sentence $\boldsymbol{x_c}$ is semantically similar to the original sentence $\boldsymbol{x}$ but may not have the same level of granularity in terms of the relation as $\boldsymbol{x}$. For example, an original sentence "*Paul ... is a senior research associate ... Library*" can be rewritten into "*Paul ... works for ... Library*".

To not disrupt the original corpus's data distribution, we will not use the rewritten sentence positive pairs directly in training. Instead, we repeat the above template extraction and NLI template grouping process on the rewritten sentences and then identify sentences from the same template groups.

The corresponding original sentence pairs can be used as additional positive pairs. Note that the ChatGPT-discovered cross-sentence positive pairs can be numerous and potentially contain a higher level of noise. To address this, we employ random sampling to select a subset of these pairs before incorporating them into the training process.

### 4.3 Representation Learning and Clustering

#### 4.3.1 K-Means Exemplar

In this work, we mainly focus on developing effective relation representations for URE tasks. For relation clustering, we adopt a simple K-Means clustering algorithm. Following the idea of Liu et al. (2022), we implement cluster centroids of different $k$ values as different granularities of relational exemplars. These exemplars are subject to fluctuations in accordance with the parameters update of the relation encoder in each training epoch.

#### 4.3.2 Loss Function

For simplicity, let $\mathcal{P}_w$ denote the set of within-sentence positive pairs, $\mathcal{P}_c$ denote the set of cross-sentence positive pairs, and $\mathcal{P}_{Pair} = \mathcal{P}_w \cup \mathcal{P}_c = [(\boldsymbol{a_r}^1, \boldsymbol{p_r}^1), ..., (\boldsymbol{a_r}^M, \boldsymbol{p_r}^M)]$ be the set of relation vectors of $M$ positive pairs from the training corpus, where $(\boldsymbol{a_r}^i, \boldsymbol{p_r}^i)$ denotes the pair of relation vectors of the $i$-th positive sample.

Liu et al. (2022) randomly sample sentences from the same minibatch as the negative samples used by the NCE loss. However, this approach may be accompanied by potential issues. When the total number of relational clusters is small, employing this approach is likely to yield a substantial number of false negatives. Furthermore, relation semantics should not be regarded as "same/different" in many cases but rather as a similarity spectrum. When considering two pairs of semantically distinct relations, it is possible for one pair to exhibit a higher degree of semantic proximity compared to the other pair. Hence, we believe the NCE loss function is not suited for relation representation learning.

We propose to apply margin loss for sentence pairs. Margin loss is more robust to noise in the training data, especially the false negative issues. Moreover, it captures the spectrum property of the differences between positive and negative pairs instead of treating them solely as binary distinctions.

The loss function for the set of sentence pairs $\mathcal{P}_{Pair}$ is defined as:

$$\mathcal{L}_{Pair} = \mathcal{L}_w + \mathcal{L}_c$$
$$= \frac{1}{M} \sum_{i=1}^{M} max\{dist(\boldsymbol{a_r}^i, \boldsymbol{p_r}^i) \quad (8)$$
$$- dist(\boldsymbol{a_r}^i, \boldsymbol{n_r}^i) + \gamma, 0\},$$

where $dist(\cdot)$ represents the cosine distance function, $\boldsymbol{n_r}^i$ is a randomly sampled negative example for $\boldsymbol{a_r}^i$, and the parameter $\gamma$, referred to as the margin, is a hyperparameter.

For the K-Means exemplar, our objective is to encourage sentences belonging to the current cluster to be closer to their centroid, while simultaneously pushing sentences outside the cluster away from the current centroid. Since a sentence either belongs to a cluster or not, this is a binary relation and we use the same Exem loss as Liu et al. (2022):

$$\mathcal{L}_{Exem} = -\sum_{i=1}^{N} \frac{1}{L} \sum_{l=1}^{L} log \frac{exp(\boldsymbol{h_r^i} * \boldsymbol{e_j^l}/\tau)}{\sum_{q=1}^{c_l} exp(\boldsymbol{h_r^i} * \boldsymbol{e_q^l}/\tau)},$$
$$(9)$$

where $j \in [1, c_l]$ represents the $j$-th cluster at granularity layer $l$, $\boldsymbol{e_j^l}$ is relation feature vector of the exemplar of instance $i$ at layer $l$, and $\tau$ is a is a temperature hyperparameter (Wu et al., 2018).

Our overall loss function is defined as the addition of Pair loss $\mathcal{L}_{Pair}$ and Exem loss $\mathcal{L}_{Exem}$:

$$\mathcal{L} = \mathcal{L}_{Pair} + \mathcal{L}_{Exem} = \mathcal{L}_w + \mathcal{L}_c + \mathcal{L}_{Exem}. \quad (10)$$

During inference, we encode each sentence using the trained model to obtain its representation. Subsequently, the K-Means clustering algorithm is applied to cluster the relation representations, enabling us to predict the relation for each sentence based on the clustering results.

## 5 Experiments

### 5.1 Datasets

Following Liu et al. (2022), we adopt NYT-FB [2] and TACRED [3] (Zhang et al., 2017) datasets to train and evaluate our model. NYT-FB is a dataset generated via distant supervision that combines information from two sources: The New York Times (NYT) articles and the Freebase knowledge graph (Bollacker et al., 2008). We filter out sentences with non-binary relations the same as Hu et al.

---

[2]https://github.com/THU-BPM/HiURE/tree/master/data_sample_for_exemple
[3]https://nlp.stanford.edu/projects/tacred/

(2020), Tran et al. (2020), and Liu et al. (2022). Tran et al. (2020) raise two additional concerns that emerge when employing the NYT-FB dataset. The development and test sets of the NYT-FB dataset consist entirely of aligned sentences without any human curation and include tremendous noisy labels. In particular, about 40 out of 100 randomly sampled sentences were given incorrect relation labels. Also, the development and test sets are part of the training set, making it challenging to accurately evaluate a model's performance on unseen sentences, even under the URE setting. Hence, supplementary evaluations are carried out on the TACRED dataset, which is a large-scale supervised relation extraction dataset built via crowdsourcing. By conducting additional evaluations on the TACRED dataset, we can attain a more precise comprehension of the performance of all the models.

## 5.2 Baselines

We compare the proposed model with the following baseline methods: 1) **EType** (Tran et al., 2020) is a straightforward relation classifier that employs a one-layer feed-forward network. It uses entity type combinations as input to infer relation types. 2) **SelfORE** (Hu et al., 2020) leverages weak, self-supervised signals from BERT for adaptive clustering and bootstraps these signals by improving contextualized features in relation classification. 3) **MORE** (Wang et al., 2021) utilizes deep metric learning to extract rich supervision signals from labeled data and drive the neural model to learn relational representations directly. 4) **OHRE** (Zhang et al., 2021) effectively integrates hierarchy information into relation representations via a dynamic hierarchical triplet objective and hierarchical curriculum training paradigm. 5) **EIURE** (Liu et al., 2021) intervenes on both the entities and context to uncover the underlying causal effects of them. 6) **HiURE** (Liu et al., 2022) is the state-of-the-art method that derives hierarchical signals from relational feature space and effectively optimizes relation representations through exemplar-wise contrastive learning. To ensure fair comparisons, we use the HiURE results with the K-Means clustering algorithm as reported in the original paper.

## 5.3 Evaluation Metrics

Following previous work, we use $B^3$ (Bagga and Baldwin, 1998), V-measure (Rosenberg and Hirschberg, 2007), and Adjusted Rand Index (ARI) (Hubert and Arabie, 1985) to evaluate the effective-

ness of different methods. For all these metrics, the higher the better.

$B^3$ precision and recall measure the quality and coverage of relation clustering, respectively. $B^3$ F1 score is computed to provide a balanced clustering performance evaluation.

V-measure is another evaluation metric commonly used for assessing clustering quality. While $B^3$ treats each sentence individually, V-measure considers both the intra-cluster homogeneity and inter-cluster completeness, providing a more comprehensive evaluation of clustering performance by considering the entire clustering structure.

Adjusted Rand Index (ARI) measures the agreement degree between the clusters assigned by a model and the ground truth clusters. It ranges in $[-1, 1]$, where a value close to 1 indicates strong agreement, 0 represents a random clustering, and negative values indicate disagreement.

## 5.4 Setup

To primarily show the enhancement of relation representations, we choose the simple K-Means as the downstream clustering algorithm. Following previous work (Simon et al., 2019; Hu et al., 2020; Liu et al., 2021, 2022), the number of clusters $k$ is set to 10. Despite the fact that the ground truth relations have more than 10 classes, this choice still provides valuable insights into the models' capability to handle skewed distributions. Moreover, this enables us to conduct fair comparisons with the baseline results. More implementation details can be found in Appendix A.

## 5.5 Main Results

We conduct experiments on both NYT-FB and TACRED datasets, with the average performance and standard deviation of three random runs reported. The main results are shown in Table 1. We observe that the proposed AugURE outperforms all baseline models consistently on $B^3$ F1, V-measure F1, and ARI metrics. On the NYT-FB dataset, the proposed AugURE on average improves the $B^3$ F1 by $0.6\%$, V-measure F1 by $4.3\%$, and ARI by $2.8\%$ compared with the runner-up results; on the TACRED dataset, the improvements are more evident, with the $B^3$ F1, V-measure F1, and ARI increased by $2.4\%$, $1.9\%$, $12.4\%$, respectively compared with the runner-up results.

The aforementioned performance gains show that the proposed relation representation learning

| Dataset | Method | B$^3$ | | | V-measure | | | ARI | External Tools and Supervision |
|---|---|---|---|---|---|---|---|---|---|
| | | Prec. | Rec. | F1 | Hom. | Comp. | F1 | | |
| NYT-FB | EType (Tran et al., 2020) | 31.3±2.1 | **63.7±2.0** | 41.9±2.0 | 31.8±2.5 | 56.2±1.8 | 40.6±2.2 | 32.7±1.9 | None |
| | SelfORE (Hu et al., 2020) | 38.5±2.2 | 44.7±1.8 | 41.4±1.9 | 37.8±2.4 | 43.3±1.9 | 40.4±1.7 | 35.0±2.0 | Parameters of SDA |
| | MORE (Wang et al., 2021) | 43.8±1.9 | 40.3±2.0 | 42.0±2.2 | **40.8±2.2** | 43.1±2.4 | 41.9±2.1 | 35.6±2.1 | None |
| | OHRE (Zhang et al., 2021) | 32.7±1.8 | 60.7±2.3 | 42.5±1.9 | 34.8±2.1 | 53.9±2.5 | 42.3±1.8 | 33.6±1.8 | None |
| | EIURE (Liu et al., 2021) | **48.4±1.9** | 38.8±1.8 | 43.1±1.8 | 37.7±1.5 | 49.2±1.6 | 42.7±1.6 | 34.5±1.4 | WikiData, T5, WebNLG |
| | HiURE w. K-Means (Liu et al., 2022) | 38.7±1.0 | 44.3±0.9 | 41.4±1.2 | 37.2±1.1 | 47.0±0.8 | 41.5±1.3 | 34.3±0.9 | None |
| | AugURE (our) | 35.7±0.8 | 56.5±1.8 | **43.7±1.1** | 40.3±0.6 | **56.3±1.2** | **47.0±0.8** | **38.4±1.6** | OpenIE, NLI, ChatGPT |
| TACRED | EType (Tran et al., 2020) | 51.9±2.1 | 47.0±1.8 | 49.3±1.9 | 52.5±2.1 | 54.8±1.9 | 53.6±2.2 | 35.7±2.1 | None |
| | SelfORE (Hu et al., 2020) | 51.6±2.0 | 44.2±1.9 | 47.6±1.7 | 51.3±2.0 | 52.9±2.3 | 52.1±2.2 | 36.1±2.0 | Parameters of SDA |
| | MORE (Wang et al., 2021) | 56.9±2.2 | 44.9±1.8 | 50.2±1.8 | 56.7±1.8 | 58.1±2.3 | 57.4±2.1 | 37.3±1.9 | None |
| | OHRE (Zhang et al., 2021) | 55.2±2.1 | 48.7±1.7 | 51.8±1.6 | 55.5±1.9 | 57.3±2.1 | 56.4±1.8 | 38.0±1.7 | None |
| | EIURE (Liu et al., 2021) | **57.4±1.3** | 47.8±1.5 | 52.2±1.4 | **57.7±1.4** | 59.7±1.7 | 58.7±1.2 | 38.6±1.1 | WikiData, T5, WebNLG |
| | HiURE w. K-Means (Liu et al., 2022) | 50.3±0.8 | 46.7±1.2 | 48.4±0.9 | 51.8±1.4 | **66.2±1.5** | 58.1±1.1 | 37.8±0.8 | None |
| | AugURE (our) | 51.3±0.8 | **58.4±3.9** | **54.6±2.2** | 56.9±1.1 | 64.9±2.4 | **60.6±1.7** | **51.0±0.6** | OpenIE, NLI, ChatGPT |

Table 1: Performance of all methods on NYT-FB and TACRED datasets.

can effectively capture the relation features for better relation clustering. Furthermore, entity annotations can influence the quality of relation representations. The performance improvements of the proposed AugURE are consistent in both the distantly-annotated NYT-FB dataset and the crowdsourcing TACRED dataset, indicating that the proposed relation representation learning is robust to noisy entity labels and relations in different domains.

It is worth mentioning that despite utilizing K-Means as the clustering algorithm, our model's performance surpasses that of both the SelfORE and OHRE, which employ more advanced clustering techniques. We believe that improving the clustering algorithm would yield advantages to the overall URE performance, and we leave it for future exploration.

## 5.6 Ablation Study

To investigate the contribution of different components, we further conduct an ablation study by systematically excluding or replacing specific components, and the results are shown in Table 2.

### 5.6.1 Effectiveness of Within-Sentence Pairs Augmentation

To assess the impact of the within-sentence pairs augmentation, two experiments are conducted: 1) removing entity-augmented positive pairs from the training set (referred to as "− Entity Augmentation"); 2) removing the entire set of within-sentence positive pairs $\mathcal{P}_w$ from the training set (referred to as "− Within-Sentence Pairs"). The results reveal that removing entity-augmented positive pairs leads to notable decreases in all metrics. This indicates that without entity augmentation, the learned relation representations may rely more on matching entity names instead of context information, posing limitations on generalizability.

It is interesting that removing the set $\mathcal{P}_w$ on the TACRED dataset leads to a performance decrease but does not show apparent influences on the NYT-FB dataset. These findings show that the proposed AugURE can learn sufficient information from the diverse cross-sentence pairs, highlighting their crucial role in enhancing the model's performance.

### 5.6.2 Effectiveness of Augmentation through Cross-Sentence Pairs

To evaluate the impact of augmentation through cross-sentence pairs, two experiments are conducted. Initially, we exclude positive pairs extracted based on ChatGPT's guidance to examine their impact (referred to as "− ChatGPT Pairs"). Subsequently, the entire set of cross-sentence positive pairs $\mathcal{P}_c$ is excluded (referred to as "− Cross-Sentence Pairs"). The results demonstrate that the impact of ChatGPT can vary across different datasets, indicating a dataset-dependent nature of its influence. This observation is reasonable since the granularity of relations across different datasets can differ, resulting in varying gaps between the original sentences and the rewritten sentences.

Excluding the set $\mathcal{P}_c$ leads to significant declines in overall performance across all metrics on both datasets. This finding highlights the effectiveness of the proposed augmentation through cross-sentence pairs in enhancing diversity and attaining superior results.

### 5.6.3 Margin Loss vs. NCE Loss

To study the advantage of margin loss compared with NCE loss, we change the within-sentence positive pairs' margin loss $\mathcal{L}_w$ in Eq. (10) to be the NCE loss defined in Eq. (1). As shown in Table 2, replacing margin loss with NCE loss leads to notable decreases in all metrics on both datasets. This result implies that the NCE loss function suits poorly for relation representation learning in URE

| Dataset | Method | B³ | | | V-measure | | | ARI |
|---|---|---|---|---|---|---|---|---|
| | | Prec. | Rec. | F1 | Hom. | Comp. | F1 | |
| NYT-FB | Full Model | 35.7±0.8 | **56.5±1.8** | 43.7±1.1 | 40.3±0.6 | 56.3±1.2 | 47.0±0.8 | **38.4±1.6** |
| | − Entity Augmentation | 33.9±0.9 | 51.4±3.8 | 40.8±1.8 | 37.7±1.2 | 52.0±2.1 | 43.7±1.5 | 35.3±3.3 |
| | − Within-Sentence Pairs | **36.5±0.3** | 55.5±1.7 | **44.0±0.7** | **40.5±0.7** | **56.5±1.7** | **47.2±1.1** | 36.9±0.5 |
| | − ChatGPT Pairs | 35.4±0.8 | 52.0±1.2 | 42.1±0.8 | 39.3±1.3 | 53.0±1.4 | 45.1±1.3 | 34.9±1.4 |
| | − Cross-Sentence Pairs | 34.8±1.7 | 47.1±2.9 | 40.0±1.7 | 37.9±1.4 | 50.2±1.9 | 43.2±1.6 | 26.3±2.8 |
| | replaced with NCE loss | 33.8±1.5 | 46.4±1.1 | 39.1±1.4 | 37.2±0.8 | 49.0±1.1 | 42.3±0.9 | 30.2±2.2 |
| TACRED | Full Model | 51.3±0.8 | 58.4±3.9 | 54.6±2.2 | 56.9±1.1 | 64.9±2.4 | 60.6±1.7 | **51.0±0.6** |
| | − Entity Augmentation | 49.0±0.3 | 53.0±1.9 | 50.9±1.0 | 53.5±0.6 | 60.3±1.5 | 56.7±1.0 | 47.3±2.1 |
| | − Within-Sentence Pairs | 50.6±0.9 | 55.3±4.5 | 52.8±2.6 | 55.4±1.5 | 62.0±2.9 | 58.5±2.1 | 45.6±3.3 |
| | − ChatGPT Pairs | **51.7±0.4** | **59.9±1.3** | **55.5±0.6** | **57.5±0.8** | **65.8±0.5** | **61.4±0.6** | 49.0±6.6 |
| | − Cross-Sentence Pairs | 49.2±1.2 | 54.3±0.3 | 51.6±0.6 | 54.3±1.4 | 61.5±0.7 | 57.7±1.1 | 48.5±0.8 |
| | replaced with NCE loss | 50.1±0.5 | 55.8±2.8 | 52.7±1.4 | 54.3±1.1 | 61.6±1.9 | 57.7±1.4 | 49.1±0.9 |

Table 2: Ablation Study.

tasks, highlighting the contribution of margin loss to the overall performance.

## 5.7 Evaluations of the OpenIE and NLI Performance

We also evaluate the OpenIE and NLI performance on the NYT-FB dataset. Results are summarized in Table 3 and Table 4, with further analyses presented in the subsequent paragraphs.

Regarding accuracy, among the 54457 same-template pairs, 73.2% of them are correct (belong to the same relation based on the ground truth labels). We randomly sampled and examined 30 of the 'incorrect' pairs and found that 90.0% of these pairs convey relationships similar enough to be regarded as positive pairs for the URE task. Among the 216 mutually entailed pairs based on NLI predictions, 63.4% of them are correct. We also randomly sampled 30 of the 'incorrect' pairs and found that 76.7% of them can be viewed as positive pairs.

In terms of coverage, the total number of extracted relation templates is 1210, and each relation template, on average, consists of 2.75 sentences. Only 33.2% of the sentences are covered by the extracted templates. That is, 66.8% of the sentences do not follow any relation template. Furthermore, the total number of extracted templates is much more than the number of target clusters for the URE task.

Consequently, the OpenIE and NLI method has high precision and is suitable to be used to generate high-quality positive pairs to enhance the contrastive learning process. The OpenIE and NLI method also suffers from low recall, which means it is unsuitable to directly serve as a baseline for the URE task.

| | Total | Correct |
|---|---|---|
| # same-template pairs | 54457 | 39882 (73.2%) |
| # mutually entailed pairs | 216 | 137 (63.4%) |

Table 3: The accuracy of OpenIE and NLI generated pairs on NYT-FB dataset.

| | NYT-FB |
|---|---|
| # original templates | 1210 |
| average # sentences per template | 2.75 |
| total # sentences | 10000 |
| # sentences covered by templates | 3322 (33.2%) |

Table 4: The coverage of OpenIE and NLI extracted templates on NYT-FB dataset.

## 6 Conclusion

In conclusion, this paper offers an exploration of URE from the perspective of relation representation learning. We introduce two novel augmentation techniques: within-sentence pairs augmentation and augmentation through cross-sentence pairs extraction, both of which aim to augment positive pairs, enhancing their diversity and bolstering the discriminative capabilities of contrastive learning. We also identify the limitations of NCE loss in URE tasks and propose margin loss for sentence pairs. Experiments on two datasets show the superiority of the proposed AugURE compared to competitive baselines, highlighting the efficacy of the augmentation techniques and the significance of employing margin loss. Overall, our findings advance the understanding of relation representation learning and provide valuable insights for future research and practical applications.

## Limitations

**Limitation of K-Means Clustering Algorithm**
The proposed AugURE adopts the K-Means clus-

tering algorithm, which requires a pre-defined number of relations $k$ as input. However, determining the appropriate number can be challenging and subjective. Moreover, K-Means assumes that clusters are spherical and have a similar size. This assumption may not hold true for datasets with skewed distributions, leading to suboptimal clustering results.

**Limitation of Relation Template Extraction**
The inherent nature of the relation template extraction process in the proposed AugURE may exhibit a bias towards larger and more frequently occurring relation types in the presence of skewed relation distributions. This bias may result in a decrease in performance when dealing with long-tail relations.

**Limitation of ChatGPT Prompt** ChatGPT is known to be sensitive to various prompts. In this work, our focus is primarily on the design of a functional ChatGPT prompt. We did not explore other strategies for sentence rewriting that may preserve the same level of granularity as the original sentence.

## Ethics Statement

We comply with the EMNLP Code of Ethics.

## Acknowledgments

The work was supported in part by the US National Science Foundation under grant NSF-CAREER 2237831. We also want to thank the anonymous reviewers for their helpful comments.

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

## A  Implementation Details

All experiments are conducted on two NVIDIA Tesla V100 SXM2 32 GB GPUs. In the training period, AdamW (Loshchilov and Hutter, 2019) is used to optimize the loss. The encoder is trained for 5 epochs with a $1e-5$ learning rate. The number of randomly sampled non-stop words $m$ is set to 2. The template frequency threshold $t$ is set to 4 for the NYT-FB dataset and 2 for the TACRED dataset. The ratio $r$ determining template entailment is set to 0.95. We use the development sets to grid search the margin hyperparameter $\gamma$ from $[0.25, 0.5, 0.75, 1.0]$ and find 0.75 is the optimal. When using the NCE loss function, we set the temperature parameter $\tau = 0.02$ and the number of negative examples $J = 10$.

## B  Statistics of Augmentation through Cross-Sentence Pairs Extraction

As introduced in Section 4.2.2, the set of cross-sentence positive pairs $\mathcal{P}_c$ can be divided into three categories: 1) same-template original positive pairs; 2) mutually entailed templates original positive pairs; 3) positive pairs extracted based on Chat-GPT's guidance. Table 5 presents a comprehensive overview of the extracted relation templates from both original sentences and ChatGPT-rewritten sentences. The table includes the number of templates for the NYT-FB dataset and the TACRED dataset, as well as statistics for the three categories of cross-sentence positive pairs.

|  | NYT-FB | TACRED |
|---|---|---|
| # original templates | 1210 | 266 |
| # rewritten templates | 633 | 259 |
| # same-template pairs | 54457 | 3876 |
| # mutually entailed pairs | 216 | 1318 |
| # ChatGPT pairs | 632188 | 57151 |

Table 5: The statistics of relation templates and cross-sentence positive pairs on NYT-FB and TACRED datasets.