# OpenReview forum: "Improving Unsupervised Relation Extraction by Augmenting Diverse Sentence Pairs"
_EMNLP/2023/Conference — EMNLP 2023 Main_

### Official Review · Reviewer_TtUf · 2023-08-01

**Typos Grammar Style And Presentation Improvements:** Please cite inline where appropriate,…
**Soundness:** 3

**Excitement:**

3: Ambivalent: It has merits (e.g., it reports state-of-the-art results, the idea is nice), but there are key weaknesses (e.g., it describes incremental work), and it can significantly benefit from another round of revision. However, I won't object to accepting it if my co-reviewers champion it.

**Paper Topic And Main Contributions:**

This work proposes a new method for the task of unsupervised relation extraction. In this formulation, an instance consists of a sentence and two marked related entities that are absent of the relation label. The task then deals with clustering a set of instances into clusters of similar relation types. The authors propose a distant supervision method and a margin-based loss function to learn latent representations of instances. After which, it is possible to apply a simple clustering technique such as k-means over the encoded instances to retrieve the clusters.

For the distant supervision approach, the authors create a dataset consisting of pairs of similar instances using two main methods:
First, given a single instance, replace some words in the sentence or some marked entities to create a new sentence.
Secondly, using a combination of OpenIE and NLI, find similar instances by comparing templates of the form subj-predicate-object derived from the source dataset of instances.
The authors propose an additional method that employs an LLM to generate new instances.

**Reasons To Accept:**

* An interesting set of data augmentation methods that create similar pairs of instances, useful to learning representations.
  * Distant supervision is based loosely on notions of entailment and building simple propositions from semantic templates.

**Reasons To Reject:**

While the method shows good results compared to baselines, and there are sufficient ablation studies, I found the method description and overall analysis to be lacking and not convincing.

1. Regarding the single instance augmentation: it is not clear what distribution words are resampled from and whether the augmented sentences remain grammatical or even make sense, or do they still carry the same relation type as their source sentence.

2. Regarding the paired sentences found through an OpenIE and NLI: while the approach is reasonable and interesting, there is no discussion or analysis of what kind of examples or relations it judges as similar, and no analysis of the mistakes it makes or what it misses. Can this OpenIE and NLI method already serve as a baseline method even without learning the underlying representations?  I believe this part is the most interesting part of the work and that the reader would be served better if they can delve deeper. Unfortunately, there is not even an example of the resulting sentence pairs let alone an error analysis.

3. The manuscript is very confusing regarding the instances created by ChatGPT. The prompt that is discussed in the paper simply requests a short description of the relation in the sentence, while the authors claim that ChatGPT has written a new and different example sentence.
4. There is no breakdown of the final augmented dataset. How many pairs of each type were there? It is unclear what contributes to the final augmented dataset.
5. The paper claims it created a more diverse set of positive (similar) instance pairs. This claim is not backed sufficiently as there is no suggested measure of diversity and no comparison with earlier works regarding the diversity of examples.
6. This manuscript lacks any example or chart that could guide the reader and explain qualitatively their method.

**Reproducibility:**

3: Could reproduce the results with some difficulty. The settings of parameters are underspecified or subjectively determined; the training/evaluation data are not widely available.

**Reviewer Confidence:**

4: Quite sure. I tried to check the important points carefully. It's unlikely, though conceivable, that I missed something that should affect my ratings.

---

> ### Author Rebuttal · Authors · 2023-08-27
>
> Thank you for your reviews and suggestions.
> 1. Regarding the single instance augmentation, we think that you are referring to positive pairs described in L275-296.
> This positive pair generation is a common practice used in previous work, and previous studies show that it helps to retain the local smoothness of the model (Liu et al., 2022). Therefore, we also include this procedure in the proposed method.
> 2. We evaluate the OpenIE and NLI performances on the NYT-FB dataset.
> Regarding accuracy, among the 54457 same-template pairs, 73.2% of them are correct (belong to the same relation based on the ground truth labels). We randomly sampled and examined 30 of the “incorrect” pairs and found that 90.0% of these pairs convey relationships similar enough to be regarded as positive pairs for the URE task. Among the 216 mutually entailed pairs based on NLI predictions, 63.4% of them are correct. We also randomly sampled 30 of the “incorrect” pairs and found that 76.7% of them can be viewed as positive pairs.
> In terms of coverage, the total number of extracted relation templates is 1210, and each relation template, on average, consists of 2.75 sentences. Only 33.2% of the sentences are covered by the extracted templates. That is, 66.8% of the sentences do not follow any relation template. Furthermore, the total number of extracted templates is much more than the number of target clusters for the URE task.
> Consequently, the OpenIE and NLI method has high precision yet suffers from low recall, making it unsuitable to directly serve as a baseline. We will add the details to the final paper.
> 3. As shown in paper L406-408, with the prompt and an original sentence “Paul ... is a senior research associate ... Library”, ChatGPT gives an output “Paul ... works for ... Library”. The output sentence is regarded as the rewritten sentence, which has the same pair of entities as the original sentence and a ChatGPT-summarized relation.
> 4. The statistics of the cross-sentence pairs are included in Appendix B. We also carefully evaluate the contributions of each type of pairs. Please refer to Section 5.6 and Table 2.
> 5. Previous URE methods generate the positive pairs mostly using within-sentence augmentation techniques described in Section 4.2.1, while we additionally utilize relation templates and NLI to extract cross-sentence positive pairs. These cross-sentence positive pairs express the same relationship but have different entities and contexts.
> For example, “When \<e1:PERSON\> Livingston </e1:PERSON> came to \<e2:LOCATION\> New York </e2:LOCATION> in 1999 , she stayed with an uncle in the Bedford-Stuyvesant section of Brooklyn , near Boys and Girls , which has a large Caribbean population in its student body of 3,500 .” and “By the time \<e1:PERSON\> Addison </e1:PERSON> moved to \<e2:LOCATION\> New York </e2:LOCATION> , '' Seebohm explains , '' he was 33 years old , and seen in hindsight these romantic failures suggest that his heterosexuality was beginning to falter .” form a cross-sentence positive pair (from NYT-FB).
> The encodings of the two sentences contain more discriminative information and our method creates more diverse positive pairs.
> 6. Thank you for the suggestion. We will include a framework chart in the final version of the paper.

---

### Official Review · Reviewer_PHta · 2023-08-01

**Soundness:** 4

**Excitement:**

3: Ambivalent: It has merits (e.g., it reports state-of-the-art results, the idea is nice), but there are key weaknesses (e.g., it describes incremental work), and it can significantly benefit from another round of revision. However, I won't object to accepting it if my co-reviewers champion it.

**Paper Topic And Main Contributions:**

This paper studies the Unsupervised Relation Extraction task. The authors focus on generating positive samples and exploring the loss function for contrastive learning. In positive sample generating, this paper proposes to generate within-sentence samples with random sampling and entity replacement, and also to find cross-sentence samples using OpenIE, NLI and ChatGPT. This paper also proposes to apply margin loss to sentence pairs.

**Reasons To Accept:**

Good insights on positive pair sampling and loss function exploring, which is often overlook by previous models and may provide different ways for future work.

**Reasons To Reject:**

1.	The contributions of generating positive samples and exploring loss function are not intuitively related, which makes this paper looks like a patchwork.
2.	Since the positive samples are generated with extra models (and annotations), the proposed method is hard to fairly compare with baseline methods. Moreover, at least the comparison of external tools and supervision should be added in Table 1.

**Reproducibility:**

4: Could mostly reproduce the results, but there may be some variation because of sample variance or minor variations in their interpretation of the protocol or method.

**Reviewer Confidence:**

3: Pretty sure, but there's a chance I missed something. Although I have a good feel for this area in general, I did not carefully check the paper's details, e.g., the math, experimental design, or novelty.

---

> ### Author Rebuttal · Authors · 2023-08-27
>
> We really appreciate your reviews and comments.
> 1. As mentioned in the Introduction section, both positive and negative examples are important for relation representation learning in URE tasks. When handling positive examples, we proposed several augmentation techniques to generate more diverse positive pairs. When handling negative examples, we argue that relation semantics should not be simply viewed as “same/different” but rather as a similarity spectrum. To this end, we propose to apply margin loss, which is also more robust to false negatives.
> 2. The data augmentation is part of the proposed method, and the ablation results of removing each augmentation technique are shown in Table 2. The same technique can be applied using different OpenIE methods, different NLI methods, or different text summarization methods. Comparison of external tools is beyond the scope of this paper.

---

### Official Review · Reviewer_Ydii · 2023-08-05

**Soundness:** 3

**Excitement:**

3: Ambivalent: It has merits (e.g., it reports state-of-the-art results, the idea is nice), but there are key weaknesses (e.g., it describes incremental work), and it can significantly benefit from another round of revision. However, I won't object to accepting it if my co-reviewers champion it.

**Paper Topic And Main Contributions:**

This paper proposes two methods for unsupervised relation extraction. One is data augmentation and the other is margin loss for sentence pairs in contrastive learning. The authors disclose that the traditional NCE loss function is not suitable for relation representation learning. Therefore, they propose to use a margin loss for sentence pairs. Experiments show that the proposed methods are effective.

**Reasons To Accept:**

It is interesting to identify the limitations of NCE loss in unsupervised relation extraction tasks. It provides insight into how to use contrastive loss in similar situations. Experiments also show a significant improvement by using margin loss.

**Reasons To Reject:**

1. Data augmentation methods seem a bit ad-hoc.
2. I think data augmentation can be done better with ChatGPT. Have you tried some simple methods like different prompts?

**Reproducibility:**

4: Could mostly reproduce the results, but there may be some variation because of sample variance or minor variations in their interpretation of the protocol or method.

**Reviewer Confidence:**

4: Quite sure. I tried to check the important points carefully. It's unlikely, though conceivable, that I missed something that should affect my ratings.

---

> ### Author Rebuttal · Authors · 2023-08-27
>
> Thank you for taking the time to review our work.
> 1. We aim to increase the context diversity using the augmentation technique. Under the unsupervised setting, it is non-trivial because the augmentation results need to have high precision to avoid introducing noise and semantic drifting issues. The proposed augmentation methods follow one simple assumption: if two sentences share the same or semantically similar relation templates, they express the same relation.
> 2. The goal of data augmentation using ChatGPT is to transform each original sentence (denoted as $x$) into a rewritten sentence $x_c$, which retains the identical underlying relationship as $x$ but in different contexts. We have tried several different ChatGPT prompts including “Given the context $x$, what is the relationship between $e_h$ and $e_t$?”, “Given the context $x$, what is the relationship between $e_h$ and $e_t$ (use less than 15 words)?”, and “Given the context $x$, what is the relationship between $e_h$ and $e_t$ (as short as possible)?”
> The first prompt generated paragraphs consisting of multiple sentences with many additional details, and thus cannot be used for our purposes. The second and third prompts generated a single sentence, but we regrettably found that they tended to change the granularity of the underlying relationship. One example is described in L406-408: an original sentence “Paul ... is a senior research associate ... Library” can be rewritten into “Paul ... works for ... Library”. Therefore, we used pattern extraction and NLI techniques on the rewritten sentences instead of using $(x, x_c)$ as a positive pair.
> In the paper’s Limitations section, we have acknowledged the possibility of unexplored strategies that could unlock an improved performance of data augmentation with ChatGPT.

---

### Meta-Review · Area_Chair_VrWM · 2023-09-24

**Recommendation:** 4

**Metareview:**

This paper presents a study on unsupervised relation extraction that involves better data augmentation. The authors also highlighted the inadequacy of NCE and argued that some alternative loss function may be more suitable. The reviewers largely acknowledge that the work is interesting and presents good results. Some major reservations include that the work seems to leverage external tools which may compromise the unsupervised setup, and that the two major techniques involved (loss, data augmentation) do not seem to naturally align with (or motivate) one another well.

---

### Decision · Program_Chairs · 2023-10-07

**Decision:**

Accept-Main

**Comment:**

This paper presents a study on unsupervised relation extraction that involves better data augmentation. The authors also highlighted the inadequacy of NCE and argued that some alternative loss function may be more suitable. The reviewers largely acknowledge that the work is interesting and presents good results. Some major reservations include that the work seems to leverage external tools which may compromise the unsupervised setup, and that the two major techniques involved (loss, data augmentation) do not seem to naturally align with (or motivate) one another well.